# Vibration, a treatment for migraine, linked to calpain driven changes in actin cytoskeleton

**Adriana J. LaGier** *, **Andrew Elbe, Amanda Thamke, Payton Anderson**

Department of Biology, College of Social and Natural Sciences, Grand View University, Des Moines, IA, United States of America

* alagier@grandview.edu

**Data Availability Statement:** All relevant data are within the paper and the data underlying the results presented in the study are available at OSF at https://osf.io/t64jg/.

## Abstract

Understanding how a human cell reacts to external physical stimuli is essential to understanding why vibration can elicit localized pain reduction. Stimulation of epithelial cells with external vibration forces has been shown to change cell shape, particularly in regards to structures involved in non-muscle cell motility. We hypothesized that epithelial cells respond to vibration transduction by altering proteins involved in remodeling cytoskeleton. Epithelial cells were exposed to vibration and assessed by microscopy, cytoskeletal staining, immunoblotting and quantitative RT-PCR. Here, we report that epithelial cell lines exposed to 15 minutes of vibration retract filopodia and concentrate actin at the periphery of the cell. In particular, we show an increased expression of the calcium-dependent, cysteine protease, calpain. The discovery that cell transitions are induced by limited exposure to natural forces, such as vibration, provides a foundation to explain how vibrational treatment helps migraine patients.

## Introduction

People working with vibration intensive machines, like a jack hammer, are exposed to prolonged vibration exposure. This type of stimulus can be deleterious and has been associated with Raynaud's disease and rheumatoid arthritis [1, 2]. However, brief and localized encounters with vibration is analgesic and is known to reduce pain of such things as injections [3], acute headaches [4] and hair restoration procedures [5]. Clinically, vibration is also used in managing diseases such as meibomian gland dysfunction [6], cerebral palsy [7, 8] and Parkinson's disease [9].

Although the positive impact of vibration on an organism has been investigated for over a century [10], there is limited information about how the cells that make up the organism respond to vibration. In truth, vibration is thought of as a good way to pop open a cell to get at its intracellular content [11]. However, with the recent re-emergence of mechanobiology [12], there is a new appreciation that natural forces, such as vibration, impact cells in meaningful ways. Therefore, understanding how a human cell reacts to vibration is essential to understanding why vibration can elicit changes in the whole organism.

At a cellular level, external vibration forces have been shown to trigger cell shape changes involving filopodia [13]. Filopodia are antennae-like structures assembled when a cell has the

**Funding:** The author(s) received no specific funding for this work.

**Competing interests:** The authors have declared that no competing interests exist.

urge to move [14]. Cell motility depends on actin cytoskeleton remodeling [15]. In particular, there is a need to establish filopodia whereby actin associated proteins work in coordination to initiate parallel actin bundles [16–18]. Filopodial linear growth and retraction is the most commonly observed dynamic of filopodia associated with cell movement [19].

In epithelial cells, filopodia are constructed conventionally with actin filament elongation at the tip and parallel actin bundling without actin branching in the shaft [20]. When epithelial cells contact another cell, the cells establish enough receptor-ligand interactions to retract filopodia [21]. Filopodial retraction involves actin depolymerization at the tip and disassembly of actin bundles in the shaft [22]. This type of actin filament remodeling relies on a large collection of proteins [reviewed in [23]] and proteins involved in treadmilling actin [24].

Calpain is one such actin remodeling protein and is required for cell movement [25] and filopodial dynamics [26]. Calpain's proteolytic activity has been associated with disconnecting integrin from actin filaments [27]. Integrin-containing filopodia are part of establishing mature cellular adhesions [28]. So, calpain's disruption of focal adhesions would provide an optimal environment for filopodial retraction. In this regard, the overarching hypothesis for this study is that vibration transduction induces calpain-driven cell shape changes.

## Materials and methods

### Cells

HeLa (ATCC-CCL-165), a human cervical epithelial carcinoma cell line, were grown in Dulbecco's Modified Eagle Media (DMEM) supplemented with 10% Fetal Bovine Serum and 50 U/ml penicillin, 50 μg/ml streptomycin and 0.5 μg/ml fungizone amphotericin B at 37°C under 95% humidity and 5% $CO_2$. Cultures were plated at 10,000 cells/cm$^2$ two days prior to the experiment and were approximately 75% confluent when vibrational stimulus was applied. This study did not involve human participants. The human cells used in this study are established and commercially available and as such were granted IRB approval to procure and use.

### Vibration

Cells were rinsed with and left in PBS (sans $Ca^{2+}$ and $Mg^{2+}$) and either placed for 15 minutes on a vibration resistant table [vibration (-)] or exposed to vibration stimulus with an analog vortex mixer (VWR) at 1,200 rpm [vibration (+)]. The use of PBS (-/-) limited potential addition of extracellular calcium and supplement noted in media.

A Teren VM-6370 vibrometer with touch probe was used to measure vibration every minute for 15 minutes. Vibration readings were most consistent when quantified velocity (mm/s) rather than acceleration or displacement, which were variable. Vibration resistant table produced 0 mm/s velocity and vortex set to 1,200 rpm produced 12.09 ± 1.69 mm/s velocity vibration, which was comparable to levels produced by standard equipment in environment (e.g. vibration produced by clock was 3.7 ± 0.1 mm/s and by air handler was 8.3 ± 0.1 mm/s). For comparison, 1,200 rpm was approximately equal to 20 Hz [1 rpm is equal to 0.0167 Hz; https://energyeducation.ca/encyclopedia/RPM], a vibrational stimulus similar to levels used to treat migraines [4].

Chosen vibration stimulus (1,200 rpm) was based on exploratory experiments documenting cell shape of cells left in 37°C $CO_2$ incubator (0.8 ± 0.1 mm/s), exposed to 600 rpm and exposed to 2,700 rpm. Cells left in incubator were similar to vibration (-) cells and cells exposed to 600 rpm had similar, albeit lessened, responses as those seen at 1,200 rpm. Cells exposed to 2,700 rpm (well outside the range of common vibration parameters), detached from the tissue culture vessel. Therefore, subsequent experiments set the vibrational stimulus at 1,200 rpm (12.09 ± 1.69 mm/s velocity).

## Microscopy

Micrographs were documented using CellSens imaging software using an inverted, Olympus CKX41 outfitted with phase contrast and fluorescence (excitation 470/40; emission 525/50) using an InfinityHD digital camera.

## Cell phenotype

Cells ($\geq$250) from combined field of views were counted as either rounded or not (having extended lamellipodial networks). Individual cell phenotype was quantified using NIH ImageJ software [29] to measure cell area (drawn manually around perimeter including any protrusions) for greater than or equal to 100 randomly chosen cells. Rounded phenotype was characterized by an overall reduction in cell area.

## Actin filament staining

Cells were fixed with 3.7% formaldehyde (sans methanol), permeabilized with 0.1% Triton X-100 and stained for actin filaments with 0.165 µM phalloidin-AlexaFluor488 (Molecular Probes). Filopodia length was quantified using NIH Image J software with 400x total magnification field of view calibrated to 450 microns. Filopodia ($\geq$100 with a maximum of five per cell) were delineated manually from cell body to end to measure filopodia length in microns.

## Immunodetection

For analysis of calpain-1 protein levels, cell lysates were prepared in lysis buffer (50mM Tris, 1% Triton-X100, 0.1% SDS, 0.5% deoxycholate, 150mM sodium chloride, and 1x protease inhibitor cocktail (Cell Signaling Technology, Danvars, MA). Lysates were cleared by centrifugation and protein concentrations were determined using a protein assay (BioRad) according to manufacturer's instructions. Samples, 20 µg total protein, were subjected to SDS-polyacrylamide gel electrophoresis on 10% Tris-glycine gels (BioRad) and blotted onto PVDF membranes using Owl semi-dry transfer blot apparatus.

Total protein present in duplicate gel was stained with Acqua Stain protein gel stain (Bulldog Bio) as per manufacturer's instructions and viewed and analyzed with BioRad Gel Doc EZ Imager Image Lab V3.0 software.

The PVDF membranes were blocked with 5% dry milk in TBS/Tween20 and then membranes were probed with 1) 1:1000 monoclonal primary antibody, mouse anti-human calpain-1 (BioRad, VMA00353) (80kD) or 2) 1:2000 monoclonal primary antibody, mouse anti-human beta-actin control (ThermoFisher, BA3R) (42kD) followed by incubation with horseradish-peroxidase (HRP)-linked goat anti-mouse secondary antibody. Colorimetric HRP substrate was visualized using Opti-4CN detection kit (BioRad) as per manufacturer's instructions. Membranes were mildly stripped (Pierce, Rockford, IL), of one set of antibodies, tested to ensure detection reagents were removed and then re-probed with the next set of antibodies. The specific band for the protein of interest was identified by its relative electrophoretic mobility (relative front) with respect to the size standard. Density of band was assigned an arbitrary volume. Specific binding was quantified by densitometry using NIH ImageJ imaging software. The protein levels were normalized to internal beta-actin. Immunoblot data represented graphically represents immunoblots performed from different experiments (n = 3).

## Calpain inhibition

Cells were pre-treated for 20 hours with DMSO solvent control [inhibitor -] or 100 µM PD150606, a calpain inhibitor (Calbiochem) [inhibitor +]. Although vibration was for only 15

minutes, calpain inhibition is reversible. So, immediately prior to exposing cells to vibration, cells were rinsed with PBS and treated with a fresh dose of DMSO or calpain inhibitor in PBS. Individual cell phenotype was quantified by cell area as described above.

## Quantitative Real-Time Reverse-Transcription Polymerase Chain Reaction (qRT-PCR)

SurePrep™ TrueTotal™ RNA Purification Kit (Fisher) was used to extract total RNA as per manufacturer's instructions. RNA quantification at $A_{260}$ used to standardize amount RNA (10 ng/μL) loaded into High-Capacity RNA to cDNA kit cDNA for reverse transcription. One-tenth of the cDNA was subjected to qRT-PCR using TaqMan® Fast Advance Master Mix (Applied Biosystems) in conjunction with primer probes (ThermoFisher): human calpain-1 (Hs00559804_m1), E-cadherin (Hs01013958_m1) and GAPDH (Hs0392907_g1) in conjunction with a QuantStudio 5 Real-Time PCR instrument. Melt curves indicate the primer set produced one product. The $\Delta\Delta C_t$ method calculation was used for comparing expression levels between cells exposed to vibrational stimulus or not.

## Statistical analysis

Two-tailed T-test (Two-sample assuming equal variances) for analysis of cell area and filopodial length, which had only two groups and ANOVA supplemented with Tukey HSD analysis was performed with Excel data analysis package.

## Results

### Cells exposed to vibration were more susceptible to losing prominent filopodia

Cells growing in non-confluent adherent conditions had prominent filopodia and lamellipodia, a characteristic of an established cell culture (**Fig 1A**). After exposure to 15 minutes of vibrational stimulus, a significant percentage of cells retracted these actin-rich plasma membrane protrusions and displayed a round shape indicative of a less adherent cell (**Fig 1B**). The number of cells displaying each cell shape was counted and calculated as a percentage of cells that were 'round'[round phenotype in $9.0 \pm 0.5\%$ cells without vibration vs $57.6 \pm 10.3\%$ cells with vibration].

Retraction of protrusions displayed by the round shaped cells resulted in a significant reduction in the average cell area of cells exposed to vibration (**Fig 1C**). Thereby, cell area was used subsequently as a quantifiable criteria for assessing round shape stimulated by vibration.

### Cells exposed to vibration have altered actin filament assembly

Filopodia are exploratory structures constructed from actin filaments [reviewed in [30]]. The retraction of filopodia indicated that vibration was impacting efficient actin filament polymerization and bundling, steps in filopodia dynamics [reviewed in [31]]. Cytoskeletal architecture was assessed with phalloidin staining of actin filaments, whereby straight needle-like rays indicate filaments bundled in a filopodia and clouds of homogenous intensity around a bright point indicate short filaments [32]. Cells growing with no vibration displayed rays indicative of filopodia (**Fig 2A**), while cells exposed to a vibrational stimulus displayed clouds of actin staining around the cell periphery indicative of shortened filaments (**Fig 2B**). Stress fibers are distinct from filopodia in that they traverse the entirety of the cell.

It was presumed that as actin bundles destabilize and filaments shortened, phalloidin staining density, quantified by image analysis, would increase. Cells exposed to vibrational stimulus had a decrease in average filopodial length in comparison to cells not exposed to vibration (**Fig**

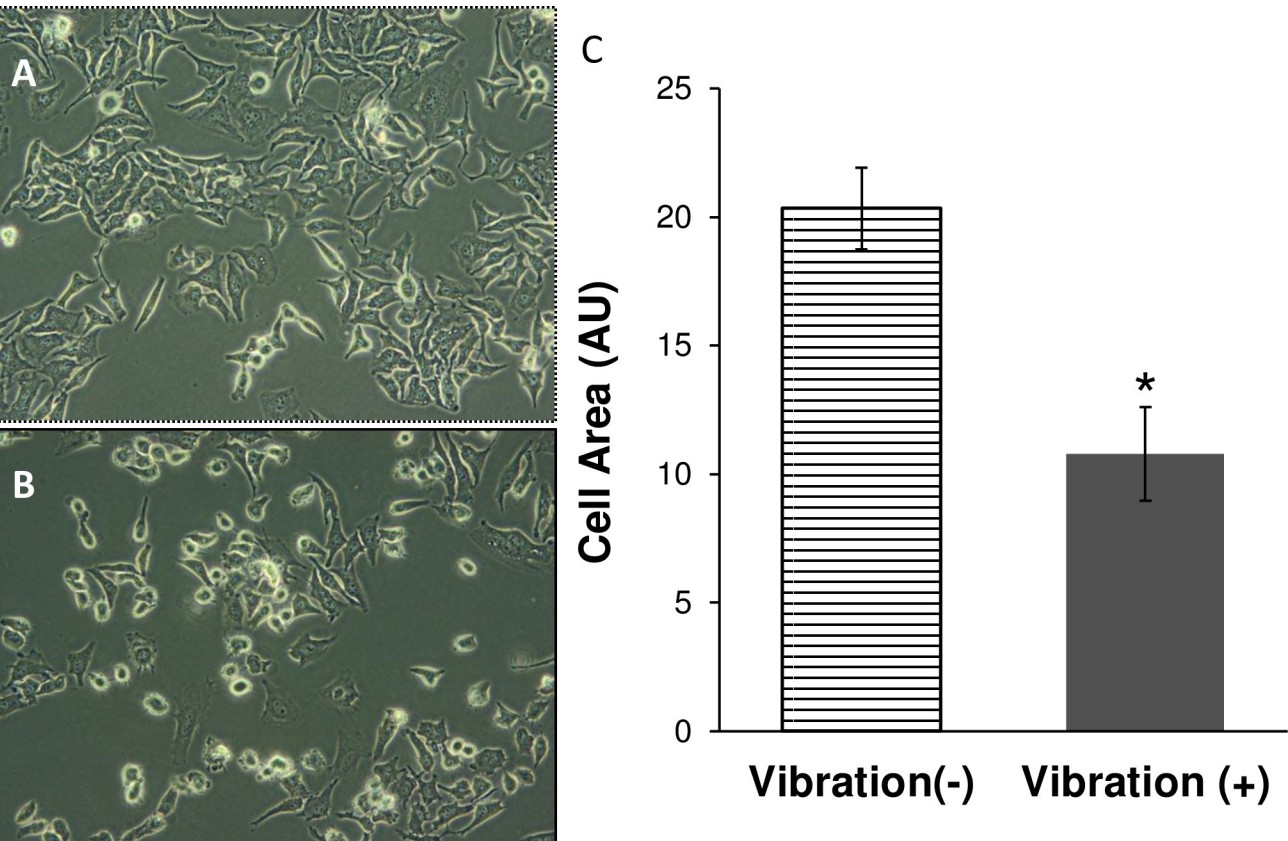

**Fig 1. Vibration induces a cell shape change consistent with loss of established filopodia.** Representative phase contrast micrographs of HeLa cells without vibration (A) and with vibration (B) at 200x total magnification. (C) The perimeter of each cell ($\geq$100) was delineated including any protrusions and the cell area was measured (arbitrary units, AU). Data shown as average cell area (AU) for all cells measured ($\pm$sem for three independent experiments). * = p < 0.05 Vibration (+) [solid bar] vs. Vibration (-) [striped bar].

**2C**). The cell staining pattern indicated a redistribution of bundled actin filaments to unbundled, shortened filaments, particularly at the cellular periphery. This pattern of staining is indicative of actin filament detachment from integrin or actin anchoring proteins.

## Vibration altered intracellular levels of protein

The results from the actin filament experiments indicated that vibration induced actin unbundling, which is mediated by actin-binding proteins and their associated proteases. To investigate whether actin associated proteins were involved, gel electrophoresis was performed on cell lysates from HeLa exposed or not exposed to vibrational stimulus (Fig 3A). An assessment of total protein relative volume (AU) revealed cells exposed to vibration had significant increases in intracellular levels of several proteins, whose band size was in the vicinity of the 72 kD protein marker (**Fig 3B**). Assessment of other bands, which showed similar or coincident decrease in protein levels (e.g. bands 1 and 6), mitigated the possibility that these alterations in proteins levels were simply a product of unequal loading of protein amounts between the samples.

## Vibration increased the intracellular level of calpain protein

A literature search on actin associated proteins with a relevant size led to calpain-1, a ubiquitous $Ca^{2+}$-activated protease. First, calpain inhibition leads to a decrease in cell detachment

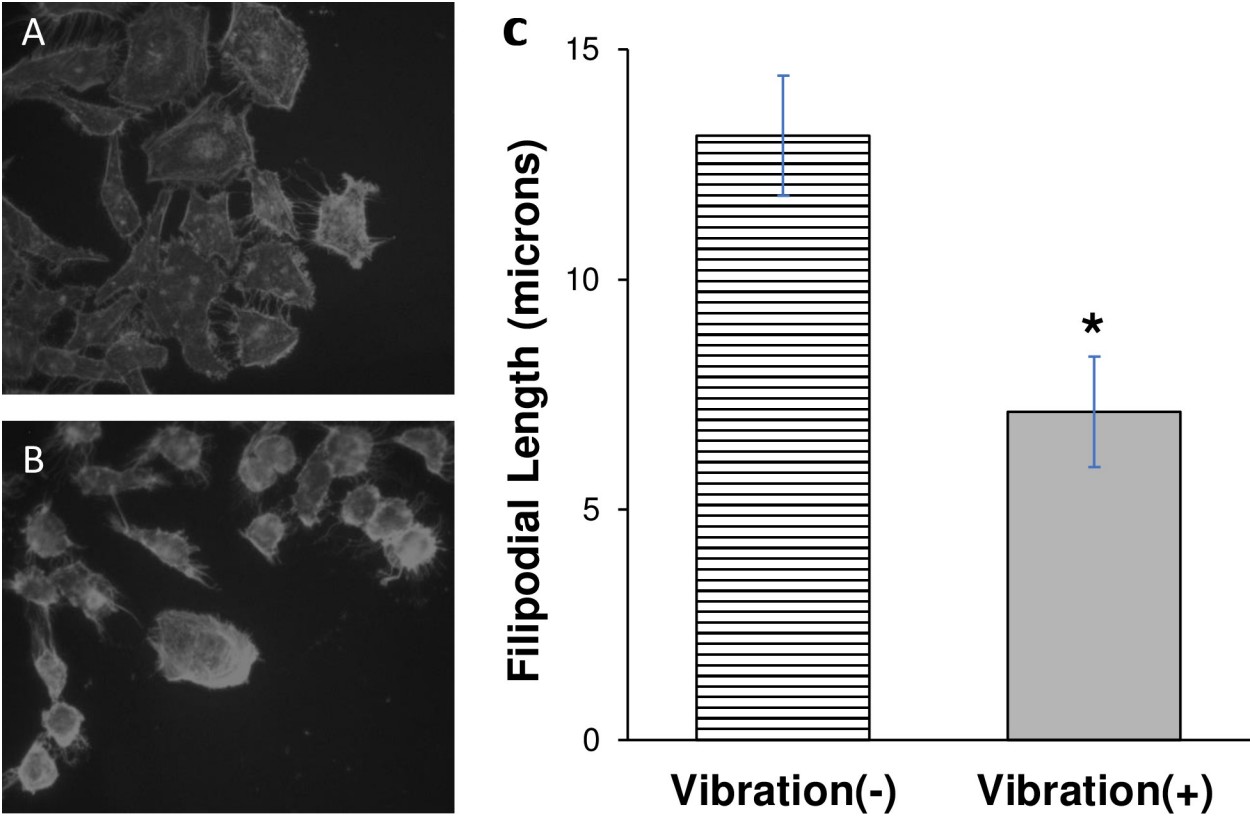

**Fig 2. Vibration alters actin filament organization.** Representative fluorescence micrographs of HeLa cells without vibration (A) and with vibration (B) stained for actin filaments. Filopodial length calculated using ImageJ image analysis software. Field of view diameter for 400x total magnification set to 450 microns. Individual filopodia ($\geq$100, with maximum of five per cell) were marked from cell body to end for length measurement. Stress fibers running across entire cell body were excluded. Data shown as average filopodial length in microns ($\pm$sem for three independent experiments) (C). * = $p < 0.05$ Vibration (+) [solid bar] vs. Vibration (-) [striped bar].

rates [25], which infers calpain involvement in increasing cell detachment or cell rounding. Second, calpain cleaves actin-associated cytoskeletal proteins [33], which infers calpain involvement in actin unbundling. Lastly, calpain-1 is 80 kD in size [34].

As determined by immunoblotting, calpain-1 levels increased as HeLa were treated with increasing vibrational stimuli. Beta-actin was detected at similar levels in each sample indicating that calpain increase was not a consequence of loading differences (**Fig 4A**). Relative calpain protein was significantly increased in cell stimulated with vibration representing 1200 rpm (V++) (**Fig 4B**). Thereby, a vibration stimulus increased calpain-1 (also known as μ-calpain) protein levels.

### Calpain inhibition attenuated vibration-induced cell shape change

Immunoblotting provides support about increases in calpain protein levels. However, it does not address calpain activity. Is calpain actively involved in cell rounding in response to a vibrational stimulus? In this regard, HeLa were pre-treated and exposed to vibration in the presence of a calpain inhibitor.

As previously documented, after exposure to 15 minutes of vibrational stimulus, a significant percentage of cells displayed a reduced cell area in comparison to cells not exposed to vibration. [note: the data presented here is independent of the data presented in previous figures]. In the presence of calpain-inhibitor a significant percentage of cells were unable to take

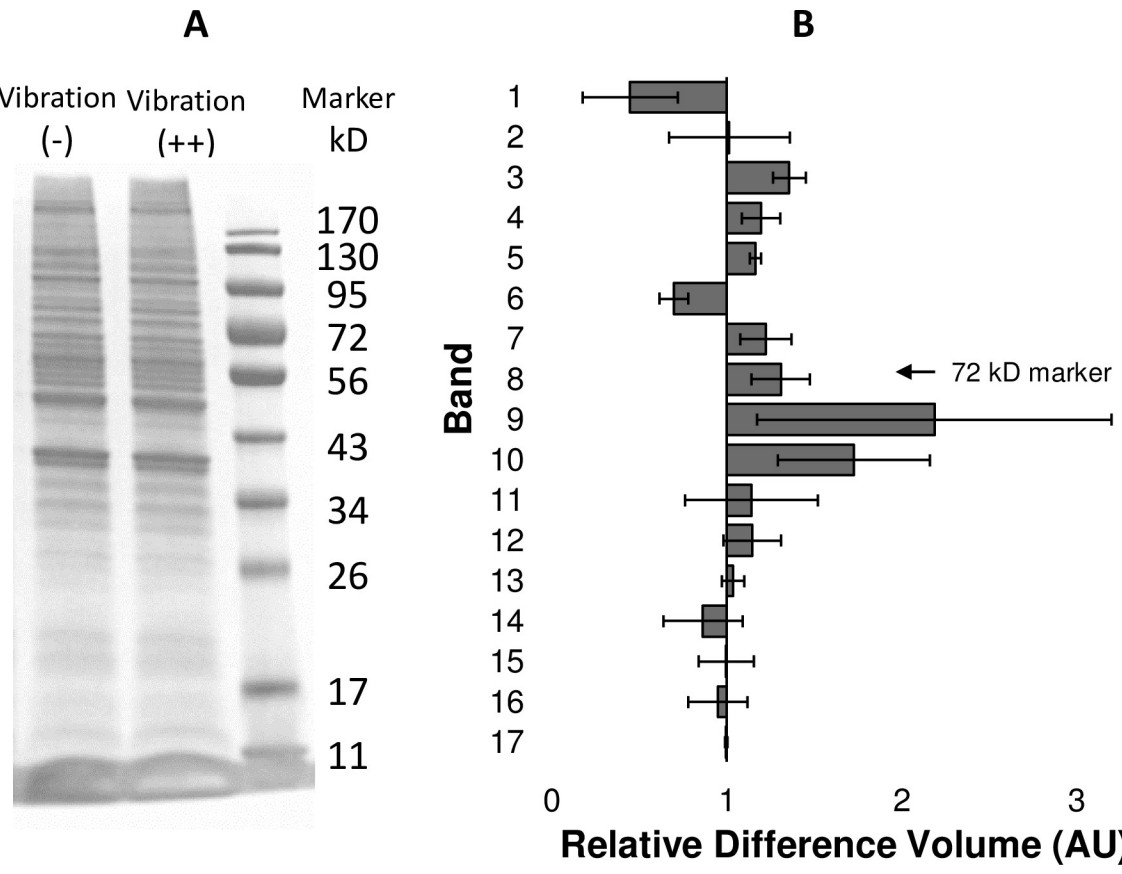

**Fig 3. Vibration altered intracellular protein content.** Representative protein bands obtained from HeLa cells without vibration and with vibration. Cells were lysed for total protein, separated by vertical electrophoresis and stained for total protein. Marker indicates 10 molecular marker bands corresponding to protein size (A). Seventeen bands from each lane were aligned based on relative front in comparison to relative front of marker bands. Arrow indicates placement of the 72kD marker. An arbitrary volume unit was assigned to each band by analysis software (B). Larger volumes were assigned as band density increased indicating an increase in protein levels. Band volumes for cells exposed to vibration are displayed as relative volume difference to cells not exposed to vibration. Numbers great than one indicate increase in vibration stimulated cells. Data shown as average relative difference volume (±sem for three independent experiments). Of note are bands 8, 9 and 10 (clustered around 72kD) that were increased in HeLa with vibration in comparison to HeLa without vibration. Alternatively, bands 1and 6 were decreased in HeLa with vibration in comparison to HeLa without vibration.

on the rounded phenotype and as such did not display the reduced cell area displayed by their counterparts that were exposed to vibration without calpain inhibitor (**Fig 5**). Therefore, vibration-stimulated changes in cell shape involve a pathway in which calpain is actively catalyzing the degradation of one of its substrates.

## Vibration altered calpain gene expression

As was noted, calpain is a ubiquitously expressed protein. It was presumed that calpain involvement in vibrational-induced cell shape changes was due to resident calpain and not as a consequence of enhanced calpain gene expression. Current trends suggest that an increase in protein levels does not necessitate an increase in gene expression [35]. However, qRT-PCR utilizing calpain-primers detected a significant fold increase in calpain expression (**Fig 6**) indicating that vibration enhances calpain gene expression.

To mitigate the possibility that vibration was inducing gene expression across the genome (rather than specifically at calpain), qRT-PCR was run on additional relevant proteins. E-

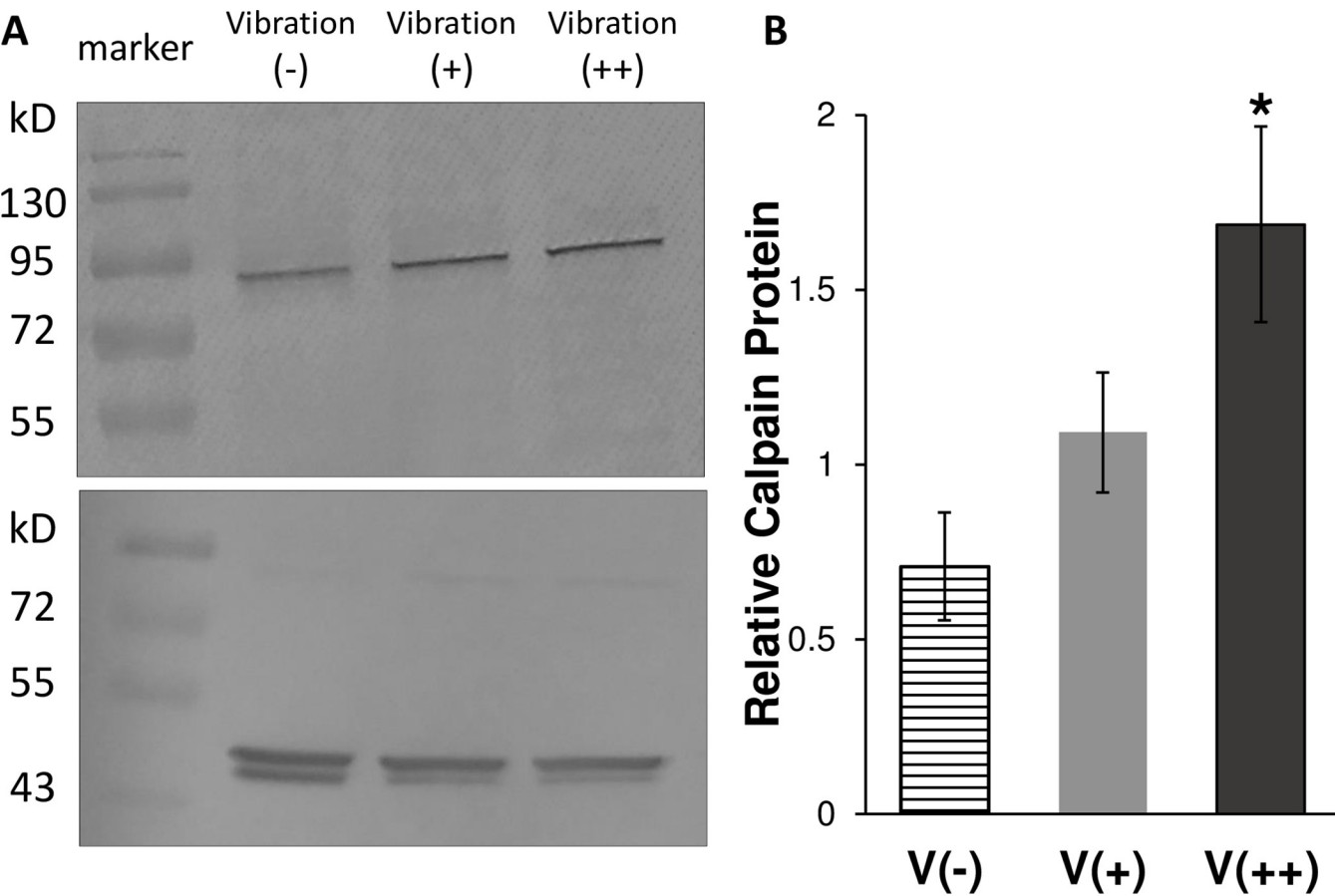

**Fig 4. Vibration stimulates increase in calpain protein.** Representative immunoblot of cytoplasmic lysates from HeLa cells without vibration (vibration -) or with increasing levels of vibration [vibration (+) and vibration (++) representing 600 and 1200 rpm, respectively]. Lysates probed for calpain-1 [top panel] and beta-actin [bottom panel] as a loading control. Similar levels of beta-actin support that calpain increase was not a consequence of loading differences. Marker indicates molecular marker bands corresponding to protein size (kD) (A). Relative calpain-1 protein, a ratio of the calpain-1 to beta-actin band peak densities for each immunoblot (n = 3), expressed as average (± sem for three independent experiments). (B). * = p < 0.05 Vibration (++) [dark, solid bar] vs. Vibration (-) [striped bar]. No significant difference noted between Vibration (+) [gray bar] vs. Vibration (-).

cadherin was chosen because enhanced calpain expression had been associated to decreased E-cadherin expression [36]. As expected, E-cadherin expression was down-regulated indicating that the enhanced calpain expression was specific to calpain.

## Discussion

The data presented here shows that epithelial cells change their shape in response to vibration. In particular, vibration stimulates a rounded shape associated with reduced cell area, actin reorganization leading to reduced filopodial length and an increase in expression of the calcium-dependent, cysteine protease, calpain-1 (μ-calpain).

Sub-confluent cultures and scratch wound assays (a portion of a confluent monolayer is removed) establish a cellular growth environment where cells crawl to find other cells. This urge to move leads to cell shape changes associated with remodeling of actin cytoskeleton [15] and is directed by antennae-like structures called filopodia [14]. The filopodia-making machine uses a convergent elongation mechanism, where filaments undergoing persistent elongation at the barbed-end eventually bundle to form the filopodia [16–18]. Retraction or

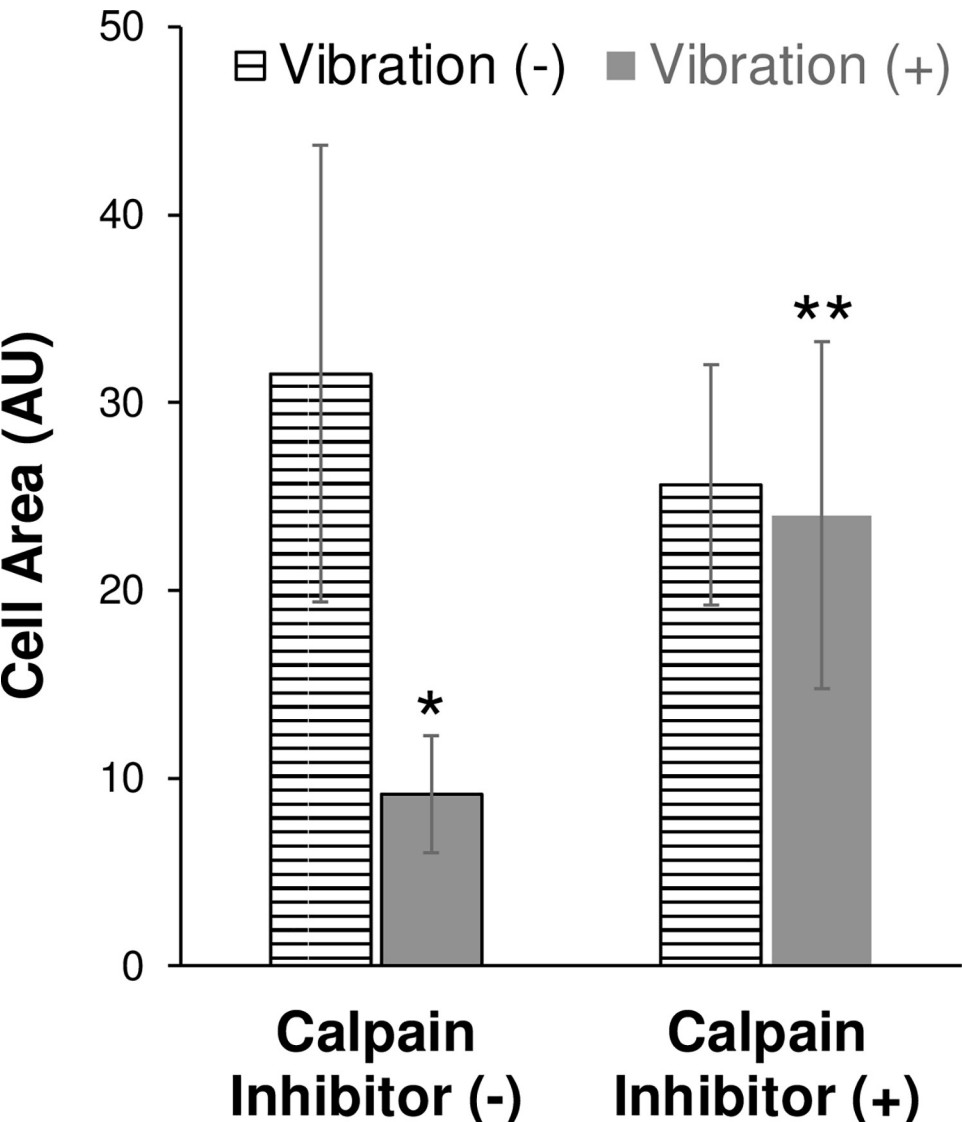

**Fig 5. Inhibition of calpain activity attenuates vibration induced cell shape change.** HeLa cells were pre-treated overnight with DMSO (solvent control) [calpain inhibitor (-)] or with calpain inhibitor, PD150606 [calpain inhibitor (+)]. HeLa cells were washed and re-treated immediately prior to treated without vibration (-) [striped bar] or with vibration (+) [solid bar]. The perimeter of each cell ($\geq$100) was delineated including any protrusions and the cell area was measured (AU). Data shown as average cell area (AU) for all cells measured (±sem for three independent experiments). * = p < 0.05 Vibration (+)/Calpain(-) [solid bar] vs. Vibration (-)/Calpain(-) [striped bar]. ** = p < 0.05 Vibration (+)/Calpain Inhibitor (+) vs. Vibration (+)/Calpain Inhibitor (-). No significant difference was noted between Vibration (+)/Calpain Inhibitor (+) and other sample groups [Vibration (-)/Calpain Inhibitor (+) or Vibration (-) / Calpain Inhibitor (-)].

loss of filopodia depends on destabilizing the barbed-end near the plasma membrane and deconstructing the filament bundles along the shaft.

Based on current findings, sub-confluent cervical epithelial cells (HeLa) cell monolayers that were exposed to vibration displayed a loss or retraction of filopodia. Vibration consistently led to cells that displayed a rounded phenotype with reduced cell area. Actin filament staining showed concentrated plaques of actin filaments around the periphery of the cell. While the remaining cells (not rounded) were observed to have a lamellipodial network consistent with shorter, branched filaments [37], which was associated with a reduction in filopodial length.

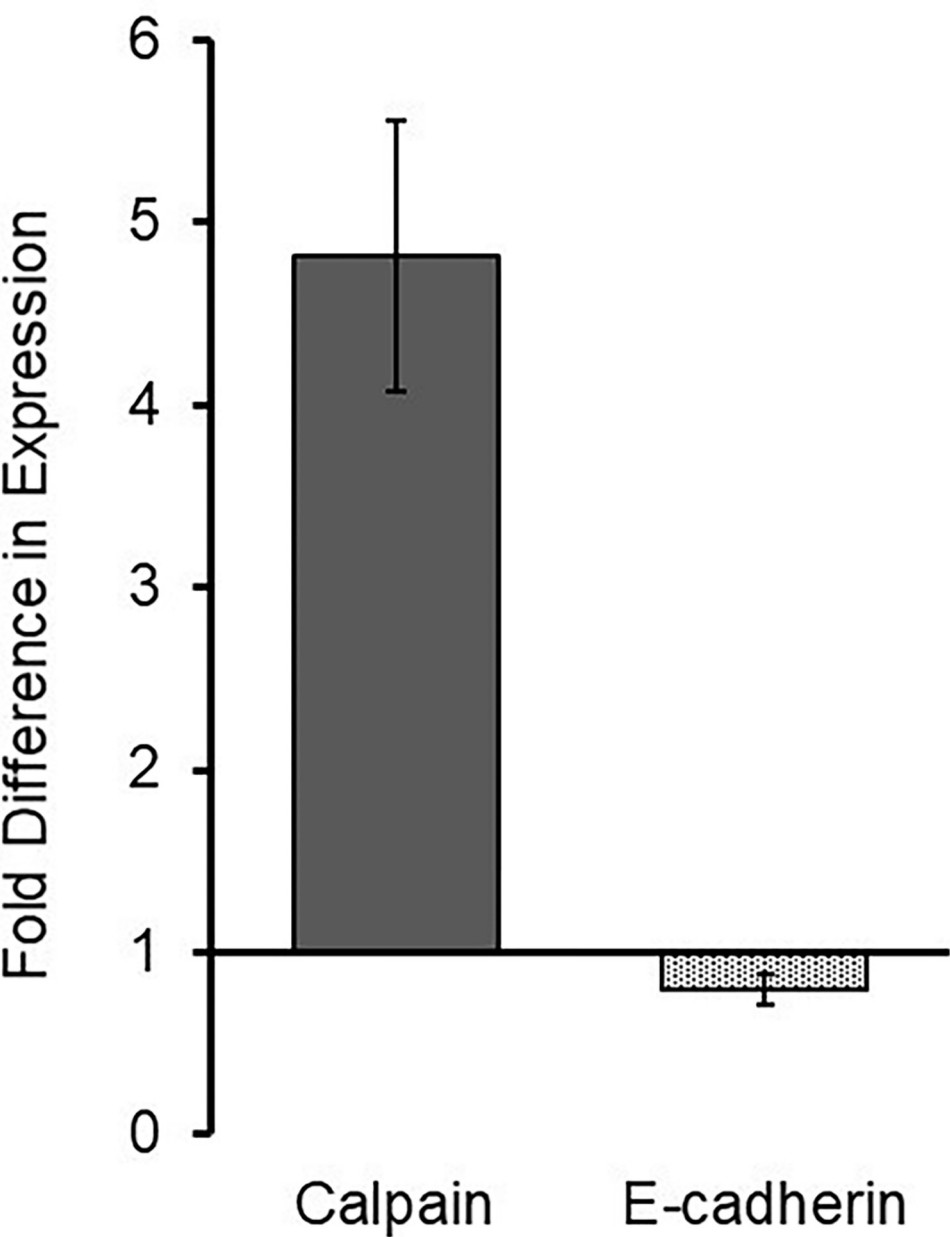

**Fig 6. Vibration alters calpain gene expression.** RNA isolated from HeLa cells without vibration or with vibration was subjected to reverse transcription. Resultant cDNA was used as a template for qPCR using primers for calpain and E-cadherin, an epithelial marker. Threshold cycles were normalized to GAPDH, a loading control. A fold difference of expression comparing vibration to no vibration of HeLa was calculated with the $\Delta\Delta C_T$ method where no change is indicated by "1". No reverse transcriptase and no template controls showed no product.

The intensity of the actin filament staining in the rounded cells was striking and was reminiscent of cells treated with cytochalasin, which disrupts actin assembly [38]. It appeared as if the filaments had collapsed or folded onto the cell, similar to plectoneme formation during nucleic acid supercoiling. This phenomenon was previously described as a mechanism to create pools of actin that could be recruited into lamellipodia [39]. The actin pools indicate depolymerization, an increase in capping and branching of the filament or a detachment from the membrane.

Congruent to the findings presented here, calpain is known to target actin membrane anchoring proteins such as vinculin in epithelial cells [40] and ezrin in platelets [41]. The data presented here shows that a 15-minute vibration exposure leads epithelial cells to increase expression of calpain-1. Furthermore, calpain activity drives vibration-induced cell rounding and associated reduction in cell area. This data together with other findings make membrane detachment of filopodial filament a feasible mechanism for how vibration induces filopodial retraction. Furthermore, observed increases in calpain expression, which could account for the significant percent change [2.7% ± 1.3] in calpain protein levels in cells with vibration vs. cells without vibration, are supported in whole animal studies where calpain mRNA increases were found in rabbit muscle tissue 45 days after exposure to hind leg vibration [42].

In this regard, calpain inhibition is expected to stabilize filopodia leading them to maintain focal contact with the plasma membrane through to the extracellular matrix at their tips. When epithelial cells were exposed to vibration in the presence of a calpain inhibitor, filopodial structures remained, which reestablished cell area similar to cells not exposed to vibration. However, others have shown in fibroblast cells that calpain inhibition actually prevented focal contact formation [33]. This paradox in reported effects of calpain inhibition has been noted previously in the literature [26] and assumedly hinges on the cell type utilized in various studies. Fibroblast-type cells, such as NIH3T3, behaving differently from epithelial cells, such as HeLa.

Clinically, vibration (kinetic oscillation) is used as an analgesic for migraine pain [4, 43]. However, there is a paucity of information to explain how vibration works at a cellular level. Others have shown that vibration reduces calcitonin-gene related peptides (CGRP) [44]. CGRP are involved in proinflammatory responses noted in migraine pathophysiology [45] and whose reduction has been shown to alleviate migraine pain in animal models [46]. CGRP increases import of calcium into cells [47, 48] indicating that reduction of CGRP would likely lead to reduction in calcium import. The data presented here indicates that vibration leads to an increase in calpain-1 levels. Calpain-1 activity is reliant on micromolar levels of intracellular calcium [49].

Together these findings imply that vibrational analgesia is a consequence of calcium signaling. This idea is supported by studies showing that mechanical stimulus leads to a significant cellular calcium response coordinating calcium influx into the cell and calcium release from the ER [50] As reported previously [51], the calpain inhibitor reported here, PD150606, works by blocking calpain's calcium binding site. As noted, this calpain inhibitor abrogated the impact of vibration lending further support to vibration mechanotransduction being driven by intracellular calcium levels. In future works, it would be interesting to perform cell fate mapping by pulsing cells with calcium to control for actin filament retraction.

The molecular processes described, such as the induction of a rounded cell shape, in conjunction with decreased expression of E-cadherin, an epithelial gene, suggests that vibration prompts the first steps of an epithelial-mesenchymal transition (EMT) [52]. Calcium signaling has also been shown to induce EMT [53]. That being said, the concept that epithelial cell transition can be mediated by cellular exposure to natural forces, like vibration, is intriguing. Moreover, the reversible nature of EMT on the onset [54] may explain why brief, localized exposure to vibration can be beneficial to human health, while prolonged exposure can be deleterious.

## Supporting information

**S1 Raw images.**
(PDF)

## Acknowledgments

Jennifer Donnelly and the Department of Biology for their dedicated support of research and Grand View University for providing and maintaining research facilities. Shena Geisinger, Sahori Ali Jaimes, Zakir Pasha and Michelle Thayer for technical assistance.

## Author Contributions

**Conceptualization:** Adriana J. LaGier, Andrew Elbe, Amanda Thamke.

**Data curation:** Adriana J. LaGier.

**Formal analysis:** Adriana J. LaGier, Andrew Elbe, Amanda Thamke, Payton Anderson.

**Funding acquisition:** Adriana J. LaGier.

**Investigation:** Adriana J. LaGier, Andrew Elbe, Amanda Thamke, Payton Anderson.

**Methodology:** Adriana J. LaGier, Amanda Thamke.

**Project administration:** Adriana J. LaGier.

**Resources:** Adriana J. LaGier.

**Supervision:** Adriana J. LaGier.

**Validation:** Adriana J. LaGier.

**Visualization:** Adriana J. LaGier.

**Writing – original draft:** Adriana J. LaGier.

**Writing – review & editing:** Adriana J. LaGier, Andrew Elbe, Amanda Thamke, Payton Anderson.

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
