## [Decision Letter · Decision Letter 0]

17 Jan 2022

PONE-D-21-39301Vibration, a treatment for migraine, linked to calpain driven changes in actin cytoskeletonPLOS ONE

Dear Dr. Lagier,

Thank you for submitting your manuscript to PLOS ONE. After careful consideration, we feel that it has merit but does not fully meet PLOS ONE’s publication criteria as it currently stands. Therefore, we invite you to submit a revised version of the manuscript that addresses the points raised during the review process. I consider that the authors should attend both reviewers recomendations and suggestions, because in its actual form the manuscript is highly descriptive and lackind statistical analysis of quantitative results.

We look forward to receiving your revised manuscript.

Kind regards,

Patricia Talamas-Rohana, Ph.D.

Academic Editor

PLOS ONE

Journal Requirements:

Additional Editor Comments:

Despite some differences in the evaluation of the two reviewers, both chose major revision for your manuscript, recognizing the relevance and interest of the topic under study.

Therefore, it is suggested to the authors to prepare a new version that addresses the recommendations of both reviewers.

Reviewers' comments:

Reviewer's Responses to Questions

**Comments to the Author**

1. Is the manuscript technically sound, and do the data support the conclusions?

Reviewer #1: Yes

Reviewer #2: Partly

2. Has the statistical analysis been performed appropriately and rigorously? 

Reviewer #1: No

Reviewer #2: Yes

3. Have the authors made all data underlying the findings in their manuscript fully available?

Reviewer #1: No

Reviewer #2: Yes

4. Is the manuscript presented in an intelligible fashion and written in standard English?

Reviewer #1: Yes

Reviewer #2: Yes

5. Review Comments to the Author

Reviewer #1: 1.- In figure 2 it is suggested to count the filopodia per cell instead of quantifying the intensity of phaloidin-488, since in the images shown by the authors, these structures are easily recognizable and quantifiable. If possible, measure the length of the filipodia in both conditions.

2.- In figure 3, it is strongly suggested to show the protein gel pattern together with the densitometry.

3.- In figure 4, how many technical replicates were made? It is necessary to do a densitometry analysis of at least two technical replicates to compare between conditions. Also, the image has no captions to recognize what conditions each line has.

4.- In all the experiments the author did not mention how many technical replicates were carried out in each experiment. This information is neccesary.

5.- In figure 1 and 5, the image shows cells with a rounded phenotype. Since this variable is presumably not qualitative, the t-student test is not adequate. Please describe in more detail how this analysis was performed or a chi-square test is suggested.

Reviewer #2: This study analyzes the impact of vibration on filopodia retraction and calpain levels in epithelial cell lines, finding that this type of stimulus induces calpain-driven changes in cell shape.

Raynaund’s disease and rheumatoid arthritis have been associated with jobs that involve the exposure to intense vibration. While, clinical vibration helps in the managing of some diseases (Meibomian gland dysfunction and Parkinson’s), and the brief and localized vibration in the analgesic effect and the reduction of some pain as acute headaches. Thus, prolonged exposure to vibration can be deleterious, while a localized exposure can be beneficial to human health. Therefore, understanding how a human cell reacts to vibration is essential to understanding why vibration can elicit changes in the whole organism either with a positive or negative effect. Then, this is an interesting study because can provide useful information about the cellular transition induced by vibration and contribute to understanding how vibrational treatments could help for pain reduction.

Specific comments:

1) The main concern is that some of the quantitative results are based on subjective observations. The percentage of cells with a rounded phenotype was determined in Figure 1. However, the shape changes are heterogeneous and the rounded phenotype criteria are not specified, such as measuring the cell area that could provide more accurate and measurable results.

2) Figure 2 shows a slight but significant difference in fluorescence intensity per cell area after vibration treatment. Changes in shape would imply a reorganization of actin cytoskeleton and not necessarily an increase in the number of actin filaments, since a greater complexity of the filaments is not observed. In this sense, it is kindly suggested that the exposure times of the captured images should be specified, since automatic image acquisition may result in different capture times and consequently give errors in fluorescence intensity measurements. If the above-mentioned problem did occur, correct accordingly.

3) The authors state that vibration altered intracellular levels of some proteins. Figure 3 shows the result of one experiment since no standard deviations are shown. A single experiment and without showing loading controls are not enough for such assertion. Therefore, the analysis of the densitometry average of at least three replicates and statistical analysis is necessary. Besides, it is kindly suggested to show the graph in a single plane to facilitate its interpretation.

4) Figure 4 is a representative immunoblot, therefore for an accurate quantification it is necessary to perform and plot the densitometry of the bands of calpain (sample) and beta-actin (loading control) of at least three replicates, normalize the results with respect to the loading control and perform the corresponding statistical analysis.

5) As in Figure 1, in Figure 5 it is necessary to define the criteria to consider when a cell is rounded, considering the heterogeneity of shapes observed. In support of this observation, in Figure 1 the percentage of cells without vibration treatment is about 7.5%, while in Figure 5 the percentage of cells in the same conditions (without exposure to vibration and without the calpain inhibitor) is about 40%. Similarly, in Figure 1 the percentage of cells treated with vibration is about 50%, while in Figure 5 the percentage of cells in the same conditions (with exposure to vibration and without the calpain inhibitor) is 75%. These discrepancies may be the result of the use of subjective criteria. Therefore, as in Figure 1 measuring cell area and set specific ranges for extended vs. rounded cell may be useful.

6) Finally, Figure 6 shows that the vibration increase calpain transcript levels (close to fivefold) suggesting an up-regulation of its gene expression. As requested above, a quantitative analysis of the results in Figure 4 and their comparison with the results in Figure 6 would provide clues whether the regulation of the calpain gene is at the transcriptional and/or translational level.

6. PLOS authors have the option to publish the peer review history of their article (what does this mean?). If published, this will include your full peer review and any attached files.

Reviewer #1: No

Reviewer #2: No

---

## [Author Response · Author response to Decision Letter 0]

2 Mar 2022

February 17, 2022RE: Response to Reviewers concerning PONE-D-21-39301

We greatly appreciate the opportunity to revise our manuscript PONE-D-21-39301 entitled “Vibration, a treatment for migraine, linked to calpain driven changes in actin cytoskeleton”.

All journal requirements have been addressed.

1) We have ensured that the manuscript has met style requirements based on provided links, particularly appropriately formatted heading and figure labels and removing funding information from acknowledgements. 2) We have matched and provided grant numbers for the award received from Carver. 3) We have removed the four mentioned ‘data not shown’ references from the manuscript as the comments did not enhance the progression of ideas being presented by the study. 4) We have included a full ethics statement in the “Methods” section of the manuscript. 5) We have provided all original, uncropped and unadjusted images mentioned in the manuscript in a public data repository that can be found at the open science framework (OSF) at the following link. https://osf.io/t64jg/?view_only=7f2c9fbd69004b4782aea68aa86fe736

Response to Reviewer’s Comments

Based on the editor’s comments, we have responded to both reviewer’s recommendations as follows.

Reviewer #1

Overall, we greatly appreciate the comments made by reviewer #1 as they have enhanced the statistical soundness and data availability of our study. As an aside, this is our first experience publishing in an open access journal and we were previously unaware of public data repositories. In this regard, we have created an open access link to our data at OSF so that all data is electronically available.

Specifically, all comments made by reviewer #1 have been addressed.

1.- In regards to the data concerning vibrational impact on actin filament assembly, particularly filopodia (figure 2), we have embraced the reviewer’s comment that these structures are easily recognizable and quantifiable. Therefore, we replaced the staining intensity data with data where we measured the length of the filipodia (as suggested). As such, we replaced figure 2C with the updated figure.

2.- In regards to the data concerning vibration altering intracellular levels of protein (figure 3), we have added figure 3A, which shows the protein gel pattern in addition to the densitometry provided in the original version.

3.- In regards to the data concerning vibration increasing intracellular level of calpain protein (figure 4), we have added figure 4B, which is a densitometry analysis for three technical replicates. We are uncertain why the image had no captions as the figure submitted had these captions. However, perhaps now that the figure has been revised to include figure 4B, these captions are evident. In addition, reference to captions are within the figure legend. Additionally, all original, uncropped and unadjusted pictures of gels with annotations are provided in the public repository.

4.- We have fixed our oversight in regards to being explicit about experimental replication. The materials and methods and figure legends now reflect how many technical replicates were performed. Additionally, data from each replicate is provided in the public repository.

5.- In regards to the statistical analysis performed on the rounded phenotype data presented in figures 1 and 5, we have taken the reviewers suggestion and enhanced the statistical analysis of this data. For the revision, we have used an Excel data analysis package with XReal stats, to perform a one-factor ANOVA supplemented with a Tukey HSD post-hoc analysis within and between groups. The method for this analysis with the addition of technical replicates was updated in the materials and methods. As an aside, based on reviewer #2 comments, the ‘rounded phenotype’ counts were replaced with cell area. 

We have revised the manuscript to include all of reviewer #1 comments and appreciate the comments which we believe have enhanced the statistical robustness and data availability of the manuscript.

Reviewer #2

Overall, we greatly appreciate reviewer #2’s comments that this study is of interest and could make a worthy contribution to understanding vibrational treatments. The comments concerning how to improve our subjective observations has substantially improved the quality of this work and we value the reviewer’s insight.

Specifically, all comments made by reviewer #2 have been addressed.

1) In regards to our observation that cells exposed to vibration were more susceptible to losing prominent filopodia (Figure 1), we initially assessed this phenomenon using a rounded phenotype. However, reviewer #2’s insight that cell area would be more quantifiable and less subjective was incredibly valuable. As such, we reassessed the data using cell area and have replaced the original figure 1C showing rounded phenotype with cell area data (revised Figure 1C). In addition, we have added a section “cell phenotype” to the materials and methods to clearly provide the criteria used to assess the ‘round phenotype’. We believe that this has provided more accurate and measurable results.

2) In regards to the data concerning vibrational impact on actin filament assembly, particularly filopodia (figure 2), we noted the reviewer’s comment that the fluorescence intensity per cell area would depend on image acquisition and implies an increase in the number of actin filaments. As we did not suppose that there was an increase of actin filaments, but rather a rearrangement of actin filaments corresponding with a retraction of filopodia, we understand the reviewer’s concerns. As such and based on a suggestion made by reviewer #1 that filopodia were easily recognizable, we have reassessed our data set by measuring filopodial length rather than fluorescence intensity. Therefore, we replaced the staining intensity data with data where we measured the length of the filipodia (replaced figure 2C with the updated figure). In addition, we added a section to the materials and methods to describe this process. We believe this has resolved the technical errors that may have accompanied our original assessment of the data. In addition, the statistical significance of the data was enhanced.

3) In regards to the statistical robustness of the data concerning vibration altering intracellular levels of protein (figure 3), we have added a representative picture of one of three gels (Figure 3A) and provided the average and error for three independent experiments (revised Figure 3 to figure 3B including error bars). The 2-D presentation of the graph was of concern because it impeded interpretation of the data. We realized that interpretation of this data would make it necessary to assess the vibration (+) band volumes in relation to the vibration (-) ones. Therefore, we graphed the data in one dimension utilizing the relative difference in volumes between vibration (+) and vibration (-). In addition, so that the gel picture and graph aligned, we converted the 2-D column graph to a 1-D bar graph. We agree with reviewer #2 that the single plane in addition to the incorporation of error bars facilitates the interpretation of data presented about intracellular levels of protein.

4) In regards to the data concerning vibration increasing intracellular level of calpain protein (figure 4), we have added figure 4B, which is a densitometric analysis of calpain levels in relation to loading controls and includes data from three independent experiments. The corresponding statistical analysis was provided in the figure legend and marked on the figure. Additionally, all original, uncropped and unadjusted pictures of the gels with annotations are provided in the public repository.

5) As in Figure 1, we have reassessed data concerning the ‘rounded phenotype’ in the absence or presence of calpain inhibitor (figure 5) utilizing the cell area criteria defined in this revision of the manuscript. The cell area criteria suggested by reviewer #2 is indeed more robust quantitatively and less subjective, as noted by the attenuated discrepancies between figure 1 and figure 5 data. As noted by the reviewer data from the original submission using the ‘rounded phenotype’ had vibration (-) cells at 7.5% rounded in figure 1 and 40% rounded in figure 5. Utilizing the cell area criteria vibration (-) cells had a 20.3 ± 1.6 area in figure 1 and 31.5 ± 12.1 area in figure 5. Additionally, data from the original submission using the ‘rounded phenotype’ had vibration (+) cells at 50% rounded in figure 1 and 75% rounded in figure 5. Utilizing the cell area criteria vibration (+) cells had a 10.8 ± 1.8 area in figure 1 and 9.1 ± 3.1 area in figure 5. As suggested by reviewer #2, we believe that the discrepancies noted in the original submission were a result of the use of a subjective criterion and that measuring cell area was useful to address these discrepancies. As such, Figure 1C and figure 5 from the original submission was replaced with Figure 1C and figure 5 in the revision which showed data using cell area.

6) In regards to data concerning vibration altering calpain gene expression, reviewer #2’s suggestion to analyze protein levels (figure 4) allowed us to compare how the increase of calpain transcript levels corresponded to an increase in protein levels. We determine that the change in protein levels was 2.7 ± 1.3 aligned well with the 4.8 ± 0.7 fold increase noted in calpain transcripts. We have added a comment to the discussion about this point.

In summation, we found great value in the reviewer’s comments. We believe that their expertise and insight has allowed us to revise the manuscript in ways that have transformed the study from being descriptive to being highly quantitative with objective criteria and robust statistical significance.

We greatly appreciate the reviewers comments that our work is relevant and an interesting topic of study because we are earnest in our belief that the study is of import.

Sincerely,

Adriana J. LaGier, Ph.D.

Associate Professor of Biology

alagier@grandview.edu

515-263-2874

---

## [Decision Letter · Decision Letter 1]

30 Mar 2022

PONE-D-21-39301R1Vibration, a treatment for migraine, linked to calpain driven changes in actin cytoskeletonPLOS ONE

Dear Dr. LAGIER,

Thank you for submitting your manuscript to PLOS ONE. After careful consideration, we feel that it has merit but does not fully meet PLOS ONE’s publication criteria as it currently stands. Therefore, we invite you to submit a revised version of the manuscript that addresses the points raised during the review process.

Both reviewers have completed their review process of the revised version of the manuscript. Both agree that the manuscript has been improved to support the conclusion by including statistical analyses of the results. However, one of the reviewers continues to make suggestions for revising the statistics performed with the results presented in Figures 1 and 2; in addition, the other reviewer has listed some minor details that you can easily correct.

We look forward to receiving your revised manuscript.

Kind regards,

Patricia Talamas-Rohana, Ph.D.

Academic Editor

PLOS ONE

Journal Requirements:

Additional Editor Comments:

Dear authors,

Both reviewers have completed their review process of the revised version of the manuscript. Both agree that the manuscript has been improved to support the conclusion by including statistical analyses of the results. However, one of the reviewers continues to make suggestions for revising the statistics performed with the results presented in Figures 1 and 2; in addition, the other reviewer has listed some minor details that you can easily correct.

Reviewers' comments:

Reviewer's Responses to Questions

**Comments to the Author**

1. If the authors have adequately addressed your comments raised in a previous round of review and you feel that this manuscript is now acceptable for publication, you may indicate that here to bypass the “Comments to the Author” section, enter your conflict of interest statement in the “Confidential to Editor” section, and submit your "Accept" recommendation.

Reviewer #1: All comments have been addressed

Reviewer #2: (No Response)

2. Is the manuscript technically sound, and do the data support the conclusions?

Reviewer #1: Yes

Reviewer #2: Yes

3. Has the statistical analysis been performed appropriately and rigorously? 

Reviewer #1: No

Reviewer #2: Yes

4. Have the authors made all data underlying the findings in their manuscript fully available?

Reviewer #1: Yes

Reviewer #2: Yes

5. Is the manuscript presented in an intelligible fashion and written in standard English?

Reviewer #1: Yes

Reviewer #2: Yes

6. Review Comments to the Author

Reviewer #1: The authors have made the suggestions according to the previous review. The article now provides the information in a clear and understandable way. However, there are some points of the statistical analysis that are still not clear. The authors mention that they did a single-factor Anova test. This test is adequate to analyze the variance for more than two groups, which is correct for figures 4 and 5. However, for figures 1 and 2 it is not clear how the analysis was performed, since the results presented only show two groups and a Anova test is not possible with these data, even some statistical programs do not allow analysis with these parameters. It is suggested to perform a Student's t-test for Figures 1 and 2.

Reviewer #2: The authors satisfactorily incorporated the requested suggestions and comments. These changes undoubtedly improved the work, taking it from a merely subjective study to a work with quantitative data and statistical significance, which supports the conclusions obtained.

Some errors were detected in the writing of the text that it is kindly suggested to correct.

1) Page 9, line 213: There is a double period at the end of the sentence: length measurement..

2) Page 12, line 264: denisties should be replaced by densities

3) Page 13, line 288: in the text between parentheses it is not understood what it means ±sem? (±sem for three independent experiments).

7. PLOS authors have the option to publish the peer review history of their article (what does this mean?). If published, this will include your full peer review and any attached files.

Reviewer #1: No

Reviewer #2: No

---

## [Author Response · Author response to Decision Letter 1]

31 Mar 2022

Journal request has been addressed. We have reviewed our reference list to ensure that it is complete and correct. We have not retracted any cited papers. However, we found two typographical errors that we have fixed (in reference number 42 the journal title listed twice and in reference number 44 the authors names were all capitals).

All Reviewers comments have been addressed. We have also addressed and revised the article based on all suggestions made by both reviewers.

In response to reviewer #1, we used a T-test, instead of an ANOVA, to analyze data presented in figures 1 and 2. As the result was the same, there were no changes made to the figures. However, we added this analysis to the materials and methods. We also updated the datasets with the new analysis at our public data repository. The new analysis (documents S1.4 Dataset Cell Shape Area Ttest and S2.5 Dataset Cell Actin Filament Ttest) can be found at the open science framework (OSF) at the following link. https://osf.io/t64jg/?view_only=7f2c9fbd69004b4782aea68aa86fe736

In response to reviewer #2, we fixed all the typographical errors mentioned and fixed two additional spacing errors.

---

## [Decision Letter · Decision Letter 2]

5 Apr 2022

Vibration, a treatment for migraine, linked to calpain driven changes in actin cytoskeleton

PONE-D-21-39301R2

Dear Dr. Adriana J. Lagier,

We’re pleased to inform you that your manuscript has been judged scientifically suitable for publication and will be formally accepted for publication once it meets all outstanding technical requirements.

Kind regards,

Patricia Talamas-Rohana, Ph.D.

Academic Editor

PLOS ONE

Additional Editor Comments (optional):

Reviewers' comments:

Reviewer's Responses to Questions

**Comments to the Author**

Reviewer #1: All comments have been addressed

2. Is the manuscript technically sound, and do the data support the conclusions?

Reviewer #1: Yes

3. Has the statistical analysis been performed appropriately and rigorously? 

Reviewer #1: Yes

4. Have the authors made all data underlying the findings in their manuscript fully available?

Reviewer #1: Yes

5. Is the manuscript presented in an intelligible fashion and written in standard English?

Reviewer #1: Yes

6. Review Comments to the Author

Reviewer #1: (No Response)

7. PLOS authors have the option to publish the peer review history of their article (what does this mean?). If published, this will include your full peer review and any attached files.

Reviewer #1: No

---

## [Editor Report · Acceptance letter]

11 Apr 2022

PONE-D-21-39301R2 

Vibration, a treatment for migraine, linked to calpain driven changes in actin cytoskeleton

Dear Dr. LAGIER:

I'm pleased to inform you that your manuscript has been deemed suitable for publication in PLOS ONE. Congratulations! Your manuscript is now with our production department. 

Kind regards, 

on behalf of

Dr. Patricia Talamas-Rohana 

Academic Editor

PLOS ONE